

# Toward on-demand measurements of greenhouse gas emissions using an uncrewed aircraft AirCore system

Zihan Zhu[1,†], Javier González-Rocha[2,†], Yifan Ding[2], Isis Frausto-Vicencio[3], Sajjan Heerah[4], Akula Venkatram[2], Manvendra Dubey[4], Don R. Collins[1], Francesca M. Hopkins[3]

[1]Center for Environmental Research and Technology, University of California, Riverside, 92521, USA
[2]Mechanicanical Engineering Department, University of California, Riverside, 92521, USA
[3]Environmental Sciences Department, University of California, Riverside, 92521, USA
[4]Earth and Environmental Sciences Division, Los Alamos National Laboratory, 87545, USA
[†]These authors contributed equally to this work.

*Correspondence to*: Javier Gonzalez-Rocha (javier.gonzalezrocha@ucr.edu)

**Abstract.** This paper evaluates the performance of a multirotor uncrewed aircraft and AirCore system (UAAS) for measuring vertical profiles of wind velocity (speed and direction) and the mole fractions of methane ($CH_4$) and carbon dioxide ($CO_2$), and presents a use case that combines UAAS measurements and dispersion modeling to quantify $CH_4$ emissions from a dairy farm. To evaluate the atmospheric sensing performance of the UAAS, four field deployments were performed at three locations in the San Joaquin Valley of California where $CH_4$ hotspots were observed downwind of dairy farms. A comparison of the observations collected on board the UAAS and an 11-m meteorological tower show that the UAAS can measure wind velocity trends with a root mean squared error varying between 0.4 and 1.1 m s$^{-1}$ when the wind magnitude is less than 3.5 m s$^{-1}$. Findings from UAAS flight deployments and a calibration experiment also show that the UAAS can reliably resolve temporal variations in the mole fractions of $CH_4$ and $CO_2$ occurring over 10 second periods or longer. Results from the UAAS and dispersion modeling use case further demonstrate that UAAS have great potential as a low-cost tool for detecting and quantifying $CH_4$ emissions in near real-time.

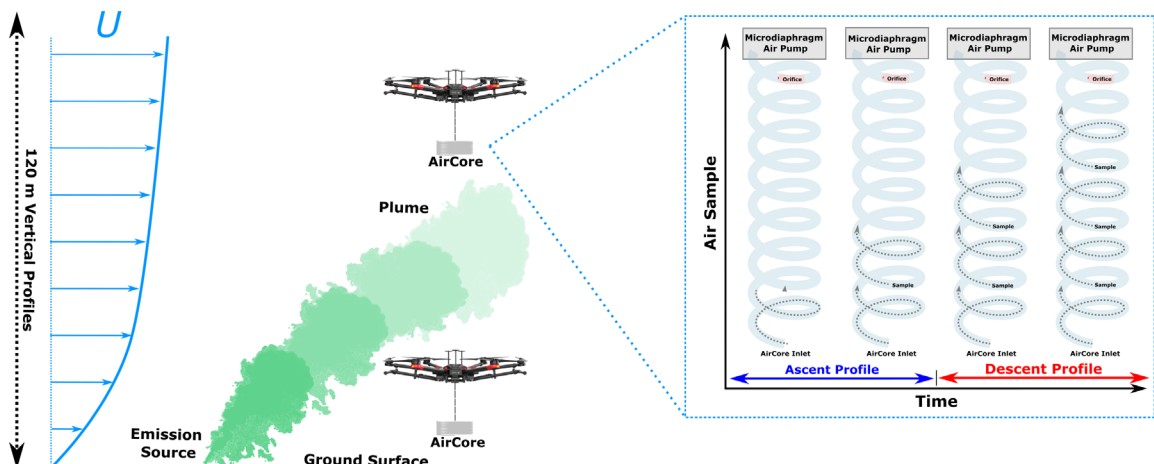



## 1 Introduction

Methane ($CH_4$) is a potent greenhouse gas responsible for a quarter of anthropogenic radiative forcing. Increases in agriculture, oil and gas, and waste management activities have contributed to the growth of atmospheric $CH_4$ levels and resultant climate warming. Due to its relatively short atmospheric lifetime of 10-12 years and high global warming potential of 84 on a 20-year timeframe (Myhre et al., 2013), $CH_4$ is an important target for climate mitigation. The Global Methane Pledge of 2021 (IEA, 2022) calls for reductions of $CH_4$ emissions, which will in turn require new measurements of baseline
emissions and verification of $CH_4$ mitigation actions. The abatement of human-driven $CH_4$ emissions will take place at individual facilities, where local $CH_4$ hotspots have been observed and emissions can be quantified, requiring further measurements to verify the success of mitigation actions.

         $CH_4$ emission estimates for individual facilities have been made through observations of wind velocity and $CH_4$ enhancements by mobile vehicle-mounted sensors, which provide the opportunity to survey a large number of facilities in
urban or agricultural settings (Moore et al., 2022; Amini et al., 2022; Arndt et al., 2018). Facilities with large emissions can be identified by atmospheric $CH_4$ enhancements adjacent to or downwind of the source observed from the ground (Hopkins et al., 2016), and then those enhancements can be converted to emission estimates with additional estimates of local winds. However, vehicle-based studies have been limited by the requirements of site or public road access, and often cannot detect emissions from elevated infrastructure such as chimneys and flare stacks. Depending on the distance of the road from the
source, the $CH_4$ plume may be lofted high above the mast of an on-road platform, particularly during daytime sampling when the planetary boundary layer height extends on the order of hundreds of meters above the surface.

         While airborne platforms do not require road accessibility and are able to provide vertical profiles of $CH_4$, airborne mass balance techniques are limited to isolated facilities in open areas (Hajny et al., 2019; Karion et al., 2013; Kobayashi et al., 2016), and are costly, which limits the potential for repeated sampling to study time-varying emissions. Plume observations
made by small uncrewed aircraft systems (sUAS) combine the flexibility of on-road measurements with the vertical profiling capabilities of aircraft. Particularly when used together with on-road sampling to identify hotspot locations, sUAS are a promising technology for facility-level methane emission estimation. Compared to light-crewed aircraft and sensor towers, small aircraft systems are low cost, portable, and can safely maneuver near emission sources at low altitudes in urban and rural environments. Such characteristics of sUAS are promising for improving the detection of $CH_4$ and $CO_2$ at sub-1km scales.
Higher-resolution observations of $CH_4$ and $CO_2$ can in turn provide more reliable estimates of anthropogenic emission sources that are difficult or infeasible to measure directly as well as detect small plumes that are not resolved by existing remote sensing technologies.

         Numerous studies have already explored the integration of low-cost sensors on board sUAS for measuring greenhouse gases. Small onboard sensors (Berman et al., 2012; Golston et al., 2017; Khan et al.,2012; Graf et al., 2018) have successfully
been used to measure multiple gas species including $CH_4$ and $CO_2$. However, low-cost $CH_4$ sensors are in an early stage of





development and do not meet the part per million (ppm) or sub-ppm sensitivity required for environmental monitoring (Honeycutt et al., 2019).

Alternate atmospheric sampling methods have combined the capabilities of multirotor sUAS and higher-precision instruments to obtain more reliable measurements of local greenhouse gas levels. For example, multiple studies have used

bag samplers for collecting lower-atmosphere air samples on board multirotor sUAS (Yuan et al., 2021; Nisbet et al., 2020; Shaw et al., 2021). Using this approach, an air volume is captured onboard the sUAS and then transported to a location where it can be analyzed using a higher-precision instrument, rendering a single point measurement for each sampling location. Other studies have aimed to obtain direct measurements of air composition by using a sUAS to tow the inlet of a high-precision instrument (Brosy et al., 2017). Although this method can increase the spatiotemporal resolution of measurements, the length

and weight of the inlet can limit air sampling operations to a small domain. Therefore, the development of unconstrained air sampling methods that can attain higher spatial and temporal resolution are necessary for accurate characterization of greenhouse gas emissions.

More practical and effective techniques for combining multirotor sUAS and high-precision air sampling instruments may be possible with AirCore technology. To date, passive and active AirCore systems have been developed and deployed on

board aircraft (Tadić and Biraud, 2018; Karion et al., 2010), weather balloons (Tu et al., 2020; Sha et al., 2020; Li et al., 2023), and sUAS (Andersen et al., 2018; Vinković et al., 2022). Passive AirCore systems rely on increases in ambient pressure for passive measurements of the atmosphere (Karion et al., 2010). Alternatively, active AirCore systems rely on a micropump and an orifice system to sample air both ascending and descending, as well as moving laterally, which provides an alternate method for increasing the spatial resolution of atmospheric measurements in the lower atmosphere. However, no study so far

has explored the integration of multirotor sUAS and AirCore systems for measuring the vertical profiles of wind velocity (i.e., wind speed and wind direction) and air composition simultaneously.

Here we evaluate the performance of a multirotor uncrewed aircraft and AirCore system (UAAS) for measuring vertical profiles of the atmospheric wind velocity and the mole fractions of $CH_4$ and $CO_2$. The UAAS was designed to measure the mole fractions of $CH_4$ and $CO_2$ in the lower 120 m of atmosphere. The motion kinematics of the UAAS were also used to

infer the wind speed and wind direction while steadily ascending and descending. The UAAS was deployed along with an on-road mobile platform to measure the mole fractions of $CH_4$ and $CO_2$ downwind of dairy farm operations. Finally, the vertical profiles of wind velocity and the mole fractions of $CH_4$ and $CO_2$ were combined with a dispersion model to detect and quantify methane emissions from a dairy farm operation. The findings from field deployments and dispersion modeling are used to assess the effectiveness of the UAAS as a low-cost solution for detecting and quantifying greenhouse gas emission sources.





## 2 Methods and materials

### 2.1 Field operations


Four UAAS operations were performed from January 20th to 24th, 2020 in the San Joaquin Valley of California to measure $CH_4$ and $CO_2$ downwind of dairy farm operations (see Table 1). $CH_4$ and $CO_2$ surveys were first conducted downwind of dairy farm facilities before each deployment (see Figure 1a) by sampling through the inlet of a Picarro G1301 cavity ring-down spectrometer (CRDS) that was placed through the side window of a van driving at a speed of approximately 32 km hr$^{-1}$.


The four UAAS deployments were performed at three locations where hot spots of $CH_4$ or $CO_2$ from dairy farms were detected (see Figure 1a). During each deployment, the UAAS profiled the wind velocity and the mole fractions of $CH_4$ and $CO_2$ steadily ascending up to a height of 120 m above ground level (AGL), and steadily descending along the same path (see Figure 1b). The air sample collected on board the UAAS was analyzed upon landing, also using the CRDS that was employed to conduct mobile surveys. The wind velocity profiles were estimated offline using the flight data collected onboard the UAAS autopilot and a kinematic vehicle motion model (González-Rocha et al., 2019a).


Table 1. Summary of UAAS flight operations conducted in the San Joaquin Valley of California.

| Location No. | Date | Pacific Standard Time | Latitude | Longitude |
|---|---|---|---|---|
| 1 | 20 January 2020 | 9:54 – 10:06 | 36°29'14.28"N | 119°21'11.88"W |
| 2 | 21 January 2020 | 15:54 – 16:05 | 36°27'49.32"N | 119°23'7.44"W |
| 2 | 21 January 2020 | 16:23 – 16:33 | 36°27'49.32"N | 119°23'7.44"W |
| 3 | 24 January 2020 | 16:38 – 16:48 | 36°28'16.68"N | 119°19'52.68"W |

### 2.2 Ground-based meteorological and gas sensors

### 2.2.1 Meteorological evaluation tower


Observations from an 11-m meteorological evaluation tower (MET) were used to assess the performance of the UAAS measuring the wind velocity trends in the lower atmosphere. The MET was located within a radius of 8.3 km from all three UAAS operations. The surface topography between the MET and the locations of the three UAAS operations was relatively flat. As shown in Figures 2a and 2b, two Campbell Scientific CSAT-3 sonic anemometers were installed on top of the MET at heights of 3 m and 11 m AGL. A CR3000 data logger was used to collect and process 1-second and 5-minute sonic anemometer measurements. The sonic anemometers and data loggers were both powered using a 12V marine deep-cycle battery.




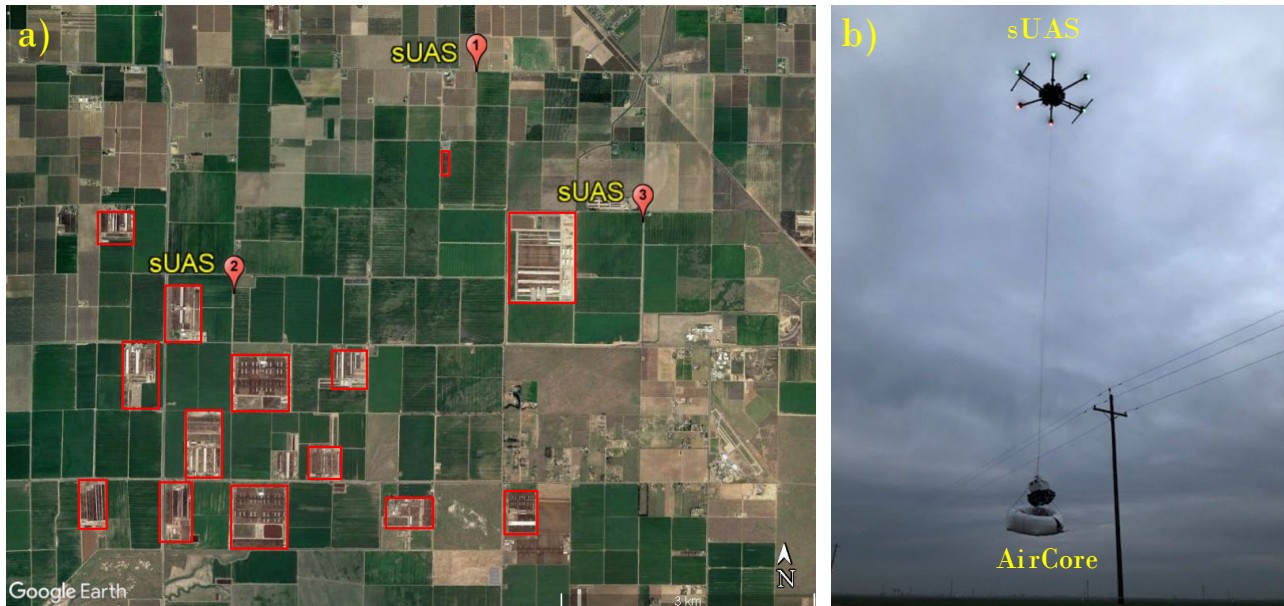

**Figure 1.** a) A satellite image from © Google Earth showing the livestock facilities surrounding the three locations where sUAAS flight operations were performed on January 20th, 21st, and 24th, 2020. b) A photo of UAAS operations near the surface.

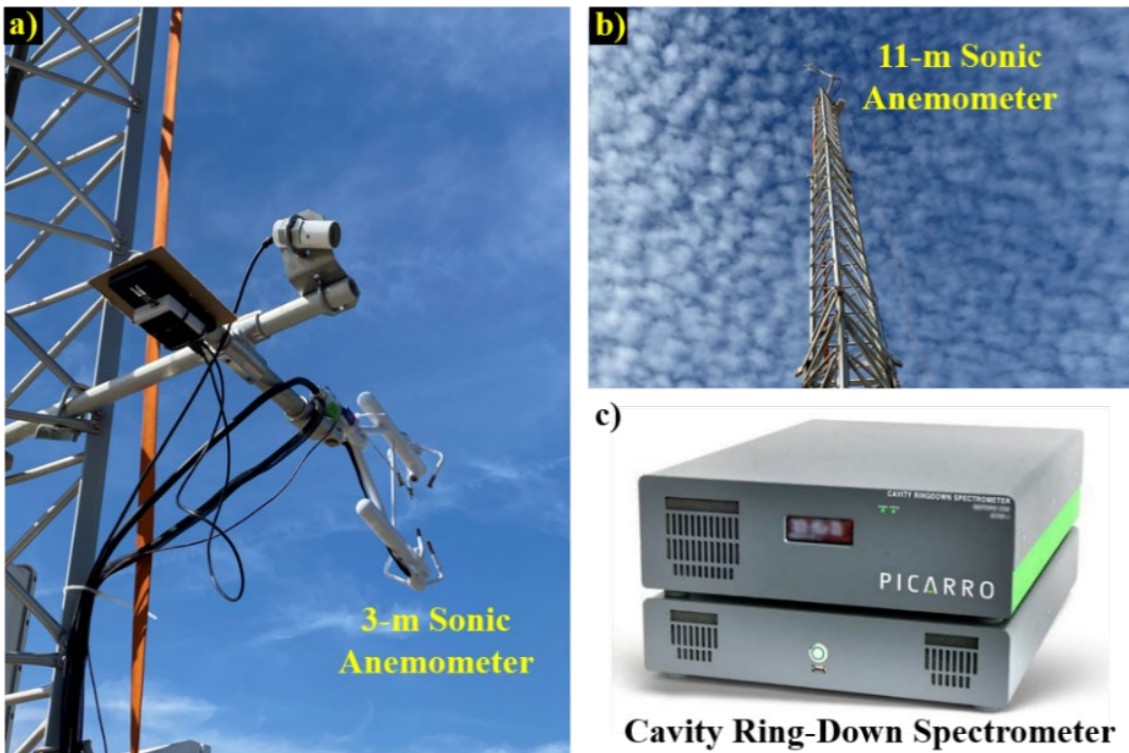


**Figure 2.** a) An image of the CSAT-3 anemometer installed on the MET tower 3 m AGL. b) An image of the CSAT-3 anemometer installed on the MET tower 11 m AGL. c) An image of the Picarro G1301 gas analyzer that was used to measure $CH_4$ and $CO_2$.



### 2.2.2. Cavity ring-down spectrometer

The CRDS that was used for field calibration and field experiments is the Picarro G1301 gas analyzer (Figure 2c).
The CRDS was housed inside a passenger van and electrical power was supplied to the instrument using a standalone 12V
marine deep cycle battery with a pure sine inverter. The instrument's flow rate was measured to be 0.7 standard liters per
minute.

### 2.3 Multirotor sUAS

The multirotor aircraft that was used to tow the AirCore system is a commercially available hexacopter Matrice 600
Pro (SZ DJI Technology, China). The Matrice 600 Pro airframe measures 1668 mm × 1518 mm × 727 mm and has a maximum
take-off payload capacity of 6 kg. The AirCore system was attached to the bottom of the multirotor airframe using a 5-m long
stainless-steel cable (see Figure 1b). Fully integrated, the UAAS has a maximum flight time of 13 minutes. The flight telemetry
record is automatically logged on-board the autopilot of the UAAS. The control and retrieval of flight records were conducted
using the DJI GO app (SZ DJI Technology, China).

### 2.4 AirCore system

### 2.4.1 Hardware description

The AirCore system consists of perfluoroalkoxy (PFA) coiled tubing that is approximately 60 m long. The coiled
tubing has an outer diameter of 12.7 mm, an inner diameter of 9.53 mm, and can hold up to 4.3 liters of air. The inlet of the
AirCore system is left open to collect ambient air. The outlet of the AirCore system is connected to a Karlsson Robotics D2028
micro diaphragm pump that weighs approximately 0.3 kg. Airflow through the AirCore system is held constant using an
O'Keefe Controls No. 9 (0.02286 cm diameter) metal orifice. The metal orifice is located 5 cm upstream of the micro
diaphragm pump as shown in Figure 3b. The pump and a 12V lithium-ion battery pack were placed in a plastic enclosure that
was positioned in the open area at the center of the AirCore coil. Fully assembled, the AirCore system weighs roughly 5 kg.

### 2.4.2 AirCore characterization experiments

Laboratory tests were conducted to evaluate the performance of the AirCore system for resolving variations in the
mole fractions of $CH_4$. We first generated a $CH_4$ mixture by diluting a $CH_4$ standard of 500 ppm inside of a Teflon bag with
room air. The Teflon bag was then connected to a three-way valve as shown in Figure 3a. The three-way valve of the calibration
apparatus was controlled to switch between the intake of the $CH_4$ mixture and the ambient air. The AirCore system and the
CRDS used in field experiments were connected using a tee junction to pull air simultaneously from the Teflon bag. During
the calibration experiment, spikes of $CH_4$ were generated for periods of 5 and 10 seconds to simulate the AirCore system
passing through a $CH_4$ plume (see Figure 3b). The CRDS provided real-time and continuous measurements of $CH_4$ while the
valve was opened and closed, and was used to analyze the air sample collected inside the AirCore system.



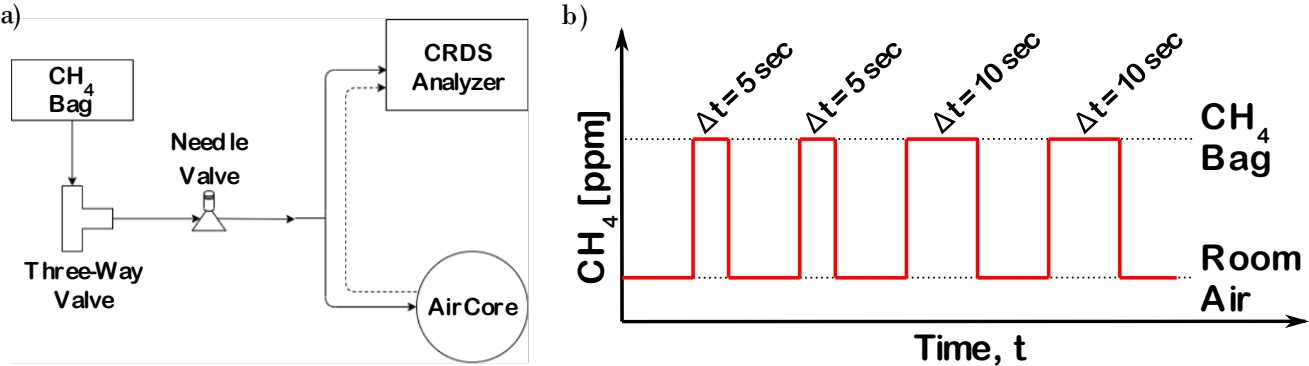

**Figure 3.** a) A schematic of the AirCore calibration experiment setup. The solid lines and arrows show the gas flow when the CRDS and AirCore sampled the $CH_4$ mixture simultaneously. The dashed lines and arrows show the gas flow when the CRDS pulls air from the AirCore system. b) A schematic showing the open and closed needle valve position during the AirCore calibration experiment.

## 2.5 Multirotor sUAS wind velocity sensing

### 2.5.1 Wind estimation method

Wind velocity profiles were estimated using a kinematic model of the UAAS and measurements of attitude and heading that were collected while steadily ascending and descending vertically using (Neumann and Bartholmai, 2015; González-Rocha et al., 2019b). The attitude and heading measurements were obtained from the attitude and heading reference system of the UAAS flight autopilot with a 20 Hz sampling rate. To develop the kinematic model, we first defined a body-fixed reference frame, $F_b = \{b_1, b_2, b_3\}$ at the aircraft center of gravity such that the unit vectors $b_1$ and $b_2$ point along the front and lateral sides of the vehicle, respectively. The unit vector $b_3$ is parallel to the propeller spin axis and points along the direction of the propulsive flow (see Figure 4a). We also defined an inertial reference frame $F_i = \{i_1, i_2, i_3\}$, affixed to the Earth's surface such that the unit vectors $i_1$ and $i_2$ point in the north and east directions, respectively, and the $i_3$ unit vector points towards the Earth's center. The orientation of the body-fixed reference frame is measured relative to the inertial reference frame using the roll-pitch-yaw Euler angles, $\Theta = \{\phi, \theta, \psi\}$. After defining the body-fixed and inertial reference frames, two kinematic relationships were derived to infer wind speed and wind direction separately.

Wind speed estimates were inferred from the tilt of the aircraft that is realized in steady-ascending vertical flight to compensate for wind disturbances. The tilt of the multirotor sUAS was determined using the dot product rule:

$$\alpha = cos^{-1}([\mathbf{R}(\phi, \theta, \psi) \cdot \mathbf{b_3}] \cdot \mathbf{i_3}) \tag{1}$$

where

$$\mathbf{R}(\phi, \theta, \psi) = \begin{pmatrix} cos\theta cos\psi & cos\psi sin\theta sin\phi - cos\phi sin\psi & cos\psi sin\theta cos\phi + sin\phi sin\psi \\ cos\theta sin\psi & sin\psi sin\theta sin\phi + cos\phi cos\psi & sin\psi sin\theta cos\phi - sin\phi cos\psi \\ -sin\theta & cos\theta sin\phi & cos\theta cos\phi \end{pmatrix} \tag{2}$$



is the rotation matrix mapping $\boldsymbol{b_3}$ from $F_b$ to $F_i$. In employing this approach, we assume there is a one-to-one relationship

between the tilt angle $\alpha$ and the horizontal wind speed (i.e., $\alpha = ||\boldsymbol{u} + \boldsymbol{v}||$ ).

The wind direction was inferred from the projection of the $\boldsymbol{b_3}$ unit vector onto the $\boldsymbol{i_1} - \boldsymbol{i_2}$ plane shown in Figure 4b during

steady-ascending flight. If the aircraft heading is pointing north, wind direction is expressed in the inertial reference frame by

computing the four-quadrant tangent inverse of the components of the $\boldsymbol{b_3}$ unit vector projected onto the $\boldsymbol{i_1}$ and $\boldsymbol{i_2}$ unit vectors

$$\beta = \tan_4^{-1}\left(\frac{[\boldsymbol{R}^T(\phi,\theta,\psi) \cdot \boldsymbol{b_3}] \cdot \boldsymbol{i_2}}{[\boldsymbol{R}^T(\phi,\theta,\psi) \cdot \boldsymbol{b_3}] \cdot \boldsymbol{i_1}}\right) \tag{3}$$

Otherwise, wind direction is expressed in the Earth-fixed reference frame by making the following correction:

$$\text{Wind Direction} = \begin{cases} \beta - \psi, & if\ \beta > \psi \\ \beta - \psi + 360, & if\ \beta < \psi \end{cases} \tag{4}$$

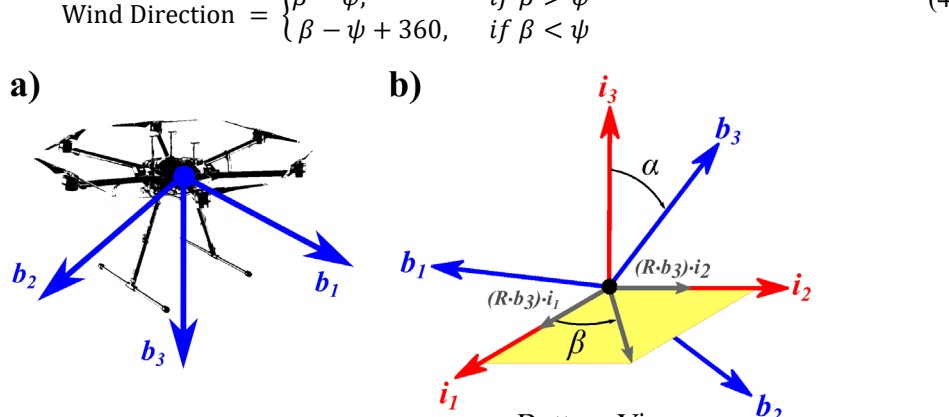

**Figure 4.** a) A schematic of the sUAS body-fixed reference frame. b) A schematic showing how the tilt angle, $\boldsymbol{\alpha}$, and wind direction, $\boldsymbol{\beta}$, are computed from the orientation of the sUAS body-fixed frame relative to the inertial reference frame.

### 2.5.2 Evaluation of Multirotor sUAS Wind Velocity Estimates

UAAS wind velocity estimates were validated employing two methods. First, we compared UAAS wind velocity

estimates to wind velocity observations collected from the 11-m MET tower described in Sect. 2.2.1. The difference between

UAAS and MET tower wind observations was quantified using the root mean squared error (RMSE) metric. Second, we

compared UAAS wind speed estimates to wind speed profiles obtained from the wind profile power law (WPPL) described in

Eq. (5)

$$U(Z_2) = U(Z_1) * \left(\frac{Z_2}{Z_1}\right)^\alpha \tag{5}$$

where $U(z_1)$ and $U(z_2)$ are the wind speeds at heights $z_1$ and $z_2$, respectively, and α is the wind shear exponent value obtained

from the MET tower at $Z_1 = 3$ m and $Z_2 = 11$ m using Eq. (6)

$$\alpha = \frac{ln\ U(z_2) - ln\ U(z_1)}{ln\ z_2 - ln\ z_1} \tag{6}$$

Results from the two assessments were used to characterize the UAAS wind estimation performance.



## 2.6 Methane emissions estimates

### 2.6.1 Dairy farm description

The vertical profiles of wind velocity and $CH_4$ collected during the first flight were used as inputs for a dispersion model to quantify $CH_4$ emissions from a dairy farm. As shown in Figure 5a, the UAAS vertical profiles were measured at a location that is 1,645 m north and 802 m west from the dairy farm, which itself is ~800 m wide. During the first UAAS operation the wind direction changed from north to south, allowing the downwind and upwind profiles to be collected from a single flight. The dairy farm in this study consists of five manure lagoons, three cattle corrals, and three cattle sheds (Figure 5b; Table 2), and is estimated to house approximately 3115 milk cows. Surface area estimates derived from Figure 5b were

used along with the UAAS vertical profiles to evaluate the effectiveness of the UAAS for estimating methane emissions.

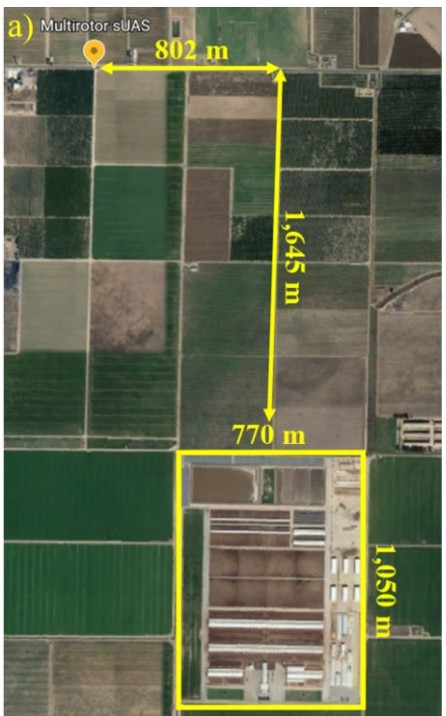
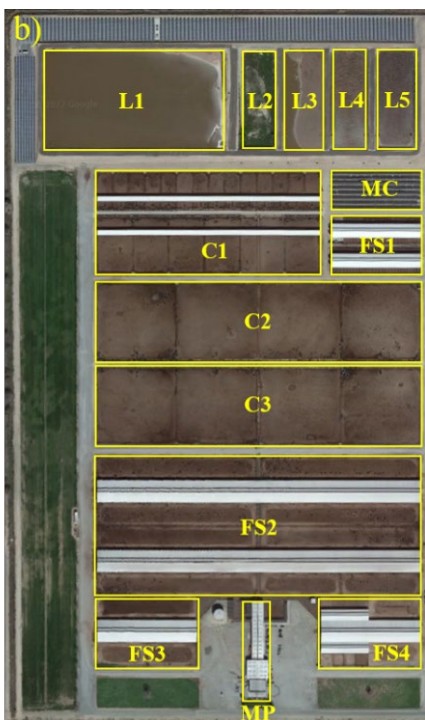

**Figure 5.** a) A satellite image from © Google Earth showing the location of the first UAAS operation downwind of a dairy farm. b) A close-up of a) showing the potential emission sources of $CH_4$ within the farm.

**Table 2**. Dairy farm sections likely to produce $CH_4$ emissions from the enteric fermentation or manure management of approximately 3115
milk cows (Marklein et al., 2021).

| Source areas | Source width | Source length | Source description |
|---|---|---|---|
| Manure lagoon | 149 m | 273 m | L1 |
| Manure lagoon | 149 m | 51 m | L2 |
| Manure lagoon | 149 m | 60 m | L3 |
| Manure lagoon | 149 m | 56 m | L4 |
| Manure lagoon | 149 m | 58 m | L5 |
| Free standing shed | 106 m | 152 m | FS1 |





| Free standing shed | 106 m | 152 m | FS2 |
|---|---|---|---|
| Free standing shed | 213 m | 494 m | FS3 |
| Free standing shed | 88 m | 137 m | FS4 |
| Cattle corral | 152 m | 342 m | C1 |
| Cattle corral | 119 m | 495 m | C2 |
| Cattle corral | 119 m | 495 m | C3 |
| Miscellaneous | 56 m | 139 m | MC |
| Milk parlor | 28 m | 145 m | MP |
| Total source area | 1,732 m | 3,049 m | TSA |

### 2.6.2 Dispersion model

The unknown emission rate from the dairy can be estimated from atmospheric observations of CH$_4$ through the following relationship:

$$C_i = \sum_{j=1}^{N} T_{ij} E_j + C_b + \varepsilon_i \tag{7}$$

where $T_{ij}$ is the transport matrix of an area source estimated by Eq. (7) with unit emission rate on data point $i$ and at source $j$,
$C_b$ is the background CH$_4$ measured from the UAAS, and $E_j$ is the inferred emission rate obtained by minimizing the residual square $\sum_i^N \varepsilon_i^2$ with the constraint that their values are greater than or equal to zero. To achieve this, we use the MATLAB function *lsqnonneg* described by Lawson and Hanson (1974). The 95% confidence intervals for the emission rate can be determined by a bootstrapping method which generates a distribution of emission rates by fitting the pseudo-observations to the model estimates.

In the numerical model, the dairy farm can be treated as an area source, which consists of a set of line sources perpendicular to the wind direction. For the contribution from each line source to the receptor, we use an analytical approximation to the integral along the source (Venkatram and Horst, 2006), which gives the concentration as

$$C(x, y, z) = q[erf(t_1) - erf(t_2)]F_z(x, z) \tag{8}$$

where

$$t_i = \frac{y - y_i}{\sqrt{2}\sigma_y x} \tag{9}$$

and $q$ is the line source emission rate per unit length, $x$ is the downwind distance of the receptor from the source, $y - y_i$ is the
distance of the receptor from two end points of the line along the direction parallel to the source, $\sigma_y$ is the horizontal plume spread, *erf* is the error function, and $F_z(x, z)$ is the vertical distribution function, which is applied the numerical solution of the mass conservation equation (Akula Venkatram and Nico Schulte, 2018)

$$U(z)\frac{\partial C}{\partial x} = \frac{\partial}{\partial z}\left(K(z)\frac{\partial C}{\partial z}\right) \tag{10}$$

where $C$ denotes the crosswind-integrated concentration $\underline{C}^y$ for convenience, $K(z)$ is the vertical eddy diffusivity, and $U(z)$ is the horizontal velocity. The boundary conditions are



$$K(z)\frac{\partial C}{\partial z} = 0 \text{ at } z = z_0 \text{ and } \frac{\partial C}{\partial z} = 0 \; at \; z = H$$

where $z_0$ is the roughness length, which is computed to be 0.005 m (Qian et al., 2010) and $H$ is the boundary layer height. The numerical method initializes a Gaussian concentration distribution at $x = 0$, which is centered at source height $z_s = 0.1$ m and with an initial vertical spread $\sigma_z = 0.1$ m. Van Ulden, (1978) shows that the analytical solution of Eq. (8) provides an excellent description of concentrations measured during the Prairie Grass Project (Barad, 1958). Venkatram (2018) evaluates the

usefulness of the analytical formulas through the numerical solution using the Businger-Dyer expressions for eddy diffusivity of heat $K_H(z)$, and the wind profile $U(z)$.

## 3 Results

Four UAAS deployments were performed at three locations in the San Joaquin Valley where $CH_4$ hotspots from dairy farms were detected. During each deployment, the UAAS measured vertical profiles of wind velocity and the mole fractions

of $CH_4$ and $CO_2$ while ascending and descending during periods of both variable and relatively-stable wind conditions. All four sets of wind velocity and air composition profiles were evaluated using ground-based measurements. The UAAS measurements and dispersion modeling were combined to estimate the methane emissions from one of the dairy farms.

### 3.1 Wind velocity profiles

The UAAS was found to reliably measure wind velocity trends while vertically ascending and descending in both

variable and relatively-stable wind conditions. The comparison of UAAS and MET wind speed observations during variable wind conditions resulted in an RMSE of 1.1 m s$^{-1}$, with the smallest error being observed while the UAAS ascended and descended near the surface (see Figure 6a). The corresponding measurements of wind direction from the UAAS and MET were in close agreement near the surface as well. In relatively-stable wind condition, the RMSE between UAAS and MET tower wind speed observations was equal to or less than 0.5 m s$^{-1}$ (see Figure 6b, 6c, and 6d). Notable differences between

UAAS and MET observations of wind direction were observed only at the start of the third flight.

The UAAS may also have good performance measuring wind speed while ascending vertically based on the evaluation of UAAS and power law wind speed profiles. As shown in Figure 7a, the uncertainty bands of the wind speed profiles that were obtained from the ascent and descent UAAS operations and the power law overlapped up to 120 m AGL and 80 m AGL, respectively, in variable winds. In relatively-stable wind conditions, the uncertainty bands of the wind speed

profiles obtained from ascent and descent operation and the power law overlapped up to 110 m AGL and 60 m AGL, respectively, as shown in Figure 7b.





**Figure 6.** A comparison of UAAS and MET wind speed observations.

**Table 3.** The integrated concentration of CRDS and AirCore measurements.

|  | CH$_4$ signal | |
| --- | --- | --- |
|  | **5 seconds** | **10 seconds** |
| CRDS [ppm] | 1,015 | 845 |
| AirCore system [ppm] | 810 | 980 |
| Percent difference | 20.2 | 16.2 |




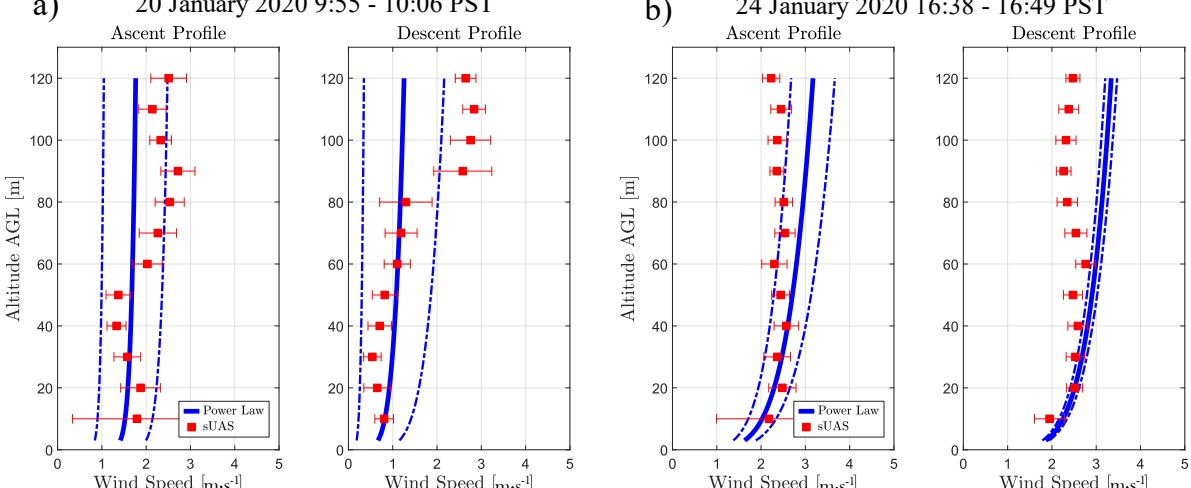

**Figure 7.** A comparison of UAAS and power law wind speed profiles measured during a) variable and b) relatively-stable wind conditions. The dashed blue lines show the uncertainty of the power law wind speed profiles.

## 3.2 $CH_4$ and $CO_2$ profiles

In addition to measuring wind velocity trends, the UAAS measured the mole fractions of $CH_4$ and $CO_2$ during both variable and relatively-stable wind conditions. As shown in Figure 8, the $CH_4$ and $CO_2$ measurements from the UAAS and those from the CRDS when sampling ambient air were in close agreement at both the start and end of the first, second, and fourth UAAS flights, which is when the UAAS and CRDS were closest to each other. During the third flight, as shown in Figure 8c, only the UAAS and CRDS measurements of $CO_2$ were found to overlap at both the start and end of the UAAS operation. The UAAS and CRDS measurements of $CH_4$ differed by 0.5 ppm at the start of the UAAS flight, which may be due to the separation between the CRDS and UAAS during deployment. Despite the measurement anomaly observed at the start, the UAAS and CRDS measurements of $CH_4$ were found to follow more consistent trends throughout the remaining period of operation.

## 3.3 AirCore characterization experiments

During laboratory characterization experiments, the UAAS was able to accurately resolve two spikes in $CH_4$ that were 10 seconds long. On the other hand, the UAAS was significantly less accurate measuring two $CH_4$ spikes that were only 5 seconds long, which is likely due to the UAAS having a slower time response. However, the lower performance of the UAAS for measuring the spikes in $CH_4$ that were 5 seconds long may have an insignificant effect on the overall reliability of $CH_4$ and $CO_2$ measurements. As shown in Table 3, the percent differences between the area under the curve of UAAS and CRD signals were found to be 20.2 and 16.2 when measuring $CH_4$ spikes that were 5 and 10 seconds long, respectively.





**Figure 8.** Comparison of UAAS and groun-level CRDS observations of $CH_4$ and $CO_2$.

### 3.4 Wind velocity and air composition profiles

As shown in Figure 9, the UAAS profiles of wind velocity and air composition were found to capture the vertical and temporal variations of $CH_4$ and $CO_2$ plumes that were measured downwind of dairy farm operations. These observations were useful for understanding how the enhancements of $CH_4$ and $CO_2$ varied in the lower atmosphere during periods of both variable and relatively-stable wind conditions.





During the first UAAS deployment, $CH_4$ and $CO_2$ plumes were observed close to the ground under variable wind conditions. The observed enhancements of $CH_4$ and $CO_2$ at low altitude are likely due to UAAS operations taking place in the morning when the boundary layer was shallow and there was minimal vertical mixing. As shown in Figure 9a, the UAAS ascent measurements captured a 1.5 ppm $CH_4$ enhancement near the ground and up to 60 m AGL. A $CO_2$ enhancement of approximately 220 ppm was observed to extend from 15 m AGL to 25 m AGL. The winds during the ascent fluctuated between

1 and 3 m s$^{-1}$ from the south, shifting eastward above a height of 117 m AGL. The descent measurements captured a 0.7 ppm enhancement of $CH_4$ extending from the ground up to 30 m AGL. The winds during the UAAS descent varied between 0.5 and 3 m s$^{-1}$ while the wind direction rotated from the east to the south and from the south to the north. $CH_4$ and $CO_2$ mole fractions were relatively constant at heights greater than 60 m AGL during the ascent and descent, indicating local background levels above the height of the dairy farm's emission plume.

The second UAAS deployment captured a methane plume moving upward during a period of turbulent conditions and relatively constant winds. As shown in Figure 9b, $CH_4$ mole fraction enhancements greater than 1 ppm were observed both at the start and end of the UAAS ascent. South-southwesterly winds were observed to gradually increase from 1.5 to 3 m s$^{-1}$ during this period. UAAS descent measurements show the mole fraction enhancement of $CH_4$ to gradually double from 1.4 to 2.8 ppm before decreasing again as south-southwesterly winds persisted. The enhancements of $CO_2$, on the other hand, were

insignificant during the UAAS ascent and descent. The observed differences between the ascent and descent measurements of $CH_4$ provide insight into the dynamic nature of plumes in well mixed afternoon conditions driven by atmospheric turbulence, which is difficult to obtain using ground-based sensing techniques alone.

      The third deployment showed an elevated $CH_4$ plume under constant wind conditions. As shown in Figure 9c, the $CH_4$ mole fraction was observed to increase from approximately 2.6 to 6 ppm both during the ascent and descent, showing the

$CH_4$ plume to vary less significantly with time. The mole fraction enhancements of $CO_2$ were greater than 50 ppm only below 40 m AGL during the descent. Southwesterly winds were observed to gradually increase with height from 1 to 2.5 m s$^{-1}$ both during the ascent and descent.

      The fourth deployment captured consistent $CH_4$ enhancements downwind of a dairy farm extending from the surface up to 80 m AGL, with a marked drop in enhancements above 100 m. $CH_4$ mole fraction enhancements of 1.7 and 1.5 ppm

were observed up to heights of 75 and 80 m AGL during the ascent and descent, respectively. The mole fraction of $CO_2$ was observed to increase briefly from 480 to 620 ppm at a height of 40 m AGL before returning to a constant value of 480 ppm for the remainder of the UAAS operation. Southeasterly winds varying between 1.5 and 3 m s$^{-1}$ persisted during both the ascent and descent. The similarity between the ascent and descent profiles measured during the fourth deployment are not surprising for the stable atmospheric conditions expected an hour before sunset.





**Figure 9.** Vertical profiles of wind velocity, $CH_4$, and $CO_2$ measured using the UAAS.

## 3.5 $CH_4$ detection and quantification

The vertical profiles of wind velocity and the mole fraction of $CH_4$ that were collected during the first UAAS operation were leveraged with the dispersion model described in Sec. 2.6.2 to detect and quantify $CH_4$ emissions from a nearby dairy

farm. We selected this set of measurements for three key reasons: 1) the nearby dairy farm is isolated from other major sources of $CH_4$, 2) the wind conditions during the UAAS operation shifted from south to north, making it possible to obtain upwind



and downwind $CH_4$ measurements from a single flight, and 3) the UAAS was able measure well above of the $CH_4$ plume to determine its height and to measure $CH_4$ background levels.

To detect $CH_4$ emissions from the nearby dairy farm, UAAS measurements of wind speed and wind direction were used with the dispersion model to generate a footprint map of $CH_4$ emissions. As shown in Figure 10, the dairy farm operation, which is denoted by a black rectangle, is well within the area having the highest contribution to the $CH_4$ mole fraction measured at the receptor (i.e., the UAAS). After generating a footprint map, UAAS profiles of wind velocity and $CH_4$ were used as dispersion modeling inputs to compute dairy farm emission estimates. Results from this analysis show that dairy farm emissions were on average 226 kg hr$^{-1}$ with a lower limit of 140 kg hr$^{-1}$ and an upper limit of 277 kg hr$^{-1}$.

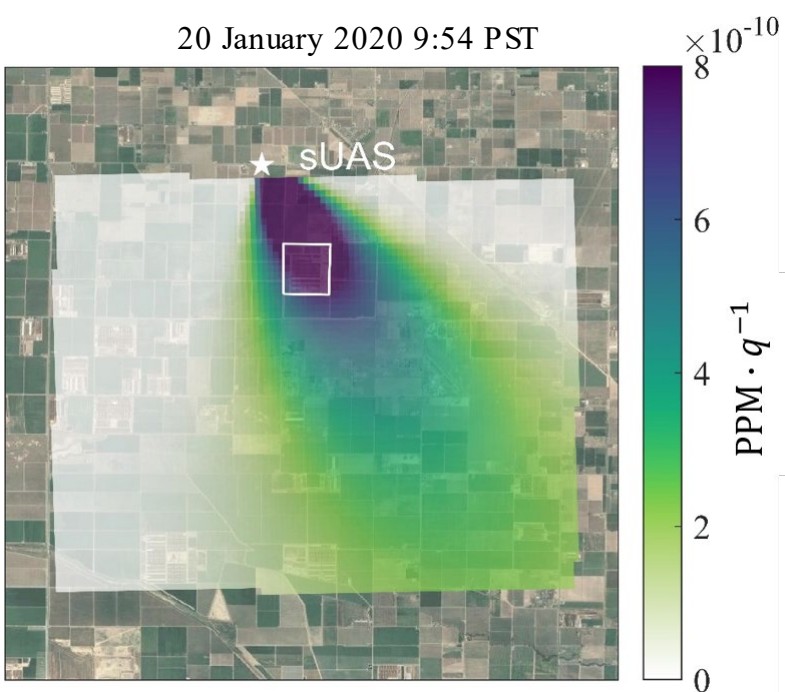


**Figure 10.** A satellite image from MathWorks overlayed with the footprint map obtained from the dispersion model. The white star shows the location of the UAAS operations, and the white rectangle shows the location of the upwind dairy farm. The color bar units are in ppm per unit of emission.

## 4 Discussion

Four UAAS deployments were successfully performed in the San Joaquin Valley to measure vertical profiles of wind velocity and the mole fractions of $CH_4$ and $CO_2$ downwind of dairy farm operations. We evaluated the reliability of the UAAS measurements using ground-based MET and CRDS observations. We also used the UAAS measurements of wind velocity and air composition to evaluate the enhancements of $CH_4$ and $CO_2$ during periods of variable and relatively-steady wind conditions. Lastly, we combined UAAS measurements and dispersion modeling as part of a use case study to determine the

utility of UAAS datasets for detecting and quantifying $CH_4$ emissions from a dairy farm operation. From this single use case



that met the requirements of emission estimation (i.e., observation of an isolated plume downwind of a nearby dairy farm where the farm falls entirely within the footprint of the observation and background $CH_4$ levels are also measured), we estimated emissions equivalent to 1.98 Gg yr$^{-1}$ (with a range of 1.23-2.43 Gg yr$^{-1}$). This range overlaps with the yearly estimated farm methane emissions of 1.44 Gg yr$^{-1}$ from a model that accounts for the number of cows and manure management practices
(Marklein et al. 2021).

We found the UAAS to provide reliable measurements of wind speed and wind direction in both variable and relatively-steady wind conditions. During variable wind conditions, the UAAS and MET observations of wind speed and wind direction were consistent when the height difference between the two systems was less than 50 m. More significant differences observed aloft between the UAAS and wind speeds derived from power law analysis of MET observations were likely due to
wind shear. During periods of relatively-stable wind conditions, the UAAS and MET observations of wind speed and wind direction were consistent up to heights of 60-80 m AGL. However, a more thorough comparison of UAAS and ground-based wind observations is required to ensure that the wind speed errors observed aloft are not the result of extrapolation errors associated with the wind profile power law.

In addition to providing observations of wind velocity, the UAAS was effective measuring the mole fractions of $CH_4$
and $CO_2$ in the lower atmosphere. UAAS and ground-level CRDS measurements of $CH_4$ and $CO_2$ were in close agreement when the UAAS operated near the surface while both ascending and descending in variable and relative-steady wind conditions. Results from the laboratory AirCore characterization experiment also demonstrated that the UAAS can accurately resolve $CH_4$ variations occurring over a period of 10 seconds or longer. Resolving variations at the 10-second scale is important for both reducing the uncertainty of $CH_4$ and $CO_2$ emission estimates and extending the spatial coverage of UAAS operations.
Furthermore, we found that the smearing effects produced by fast $CH_4$ variations led only to a 5 percent difference in the area under the curve of the 5- and 10-second long $CH_4$ signals. The latter finding suggests that smearing effects may only have a small impact in the accuracy of $CH_4$ and $CO_2$ column measurements.

Future work will address a number of limitations that were encountered while validating UAAS wind velocity measurements. First, UAAS flight operations will be conducted next to conventional *in-situ* and remote wind sensors (e.g.,
mast towers and LiDAR instruments) to better characterize the uncertainty of UAAS wind estimates. Additionally, more sophisticated dynamic models will be explored as a means to increase the accuracy of wind estimates, both ascending vertically and moving laterally. Previous studies have shown that dynamic models can render higher-fidelity wind estimates from the motion of sUAS (Gonzalez-Rocha, 2019; 2020). Higher-fidelity measurements of wind velocity and mole fraction measurements obtained from both lateral and vertical profiles will likely lead to improved emission estimates. Combined with
vertical profiles, lateral measurements of wind velocity and the $CH_4$ mole fraction can help determine the horizontal spread of $CH_4$ plumes.

Findings from the use case study show that combining UAAS measurements and dispersion modeling can help to detect and quantify $CH_4$ emission from large source areas. The region of the footprint map with the highest $CH_4$ sensitivity was found to overlap with the location of the downwind dairy farm. In addition to aiding the detection of large emission



sources, UAAS measurements and dispersion modeling can provide emission estimates in just hours. In addition to the system design issues that can be improved, these flights provide guidance for the meteorological conditions and spatial considerations for $CH_4$ emission estimation.   For example, the comparison of all four sets of profiles shown in Figure 9 suggests that measurement conditions are most favorable in the morning when the UAAS can fly well above the height of the planetary boundary layer and before the regional signals get mixed.

Overall, we found the UAAS to be a promising low-cost solution for detecting and quantifying greenhouse gas emissions from dairy farm operations. Leveraged with a ground-based mobile laboratory, the UAAS can be deployed at sites where greenhouse gas hotspots are prevalent provided that airspace access is available. The UAAS measurements of wind velocity and air composition are useful for understanding how $CH_4$ and $CO_2$ enhancements vary with height and time, providing higher resolution observations for monitoring lofted plumes. Combined with dispersion modeling, UAAS

measurements are also useful for detecting and quantifying greenhouse gas emissions from dairy farms. Furthermore, the extension of UAAS capabilities to measure horizontal transects of wind velocity and air composition can help characterize the spatial heterogeneity of large emission sources.   This capability can provide a new paradigm for improving bottom-up estimates of greenhouse gas emissions from dairy farm operations and other important sources such as waste landfills, gas and oil fields, and wetlands. Ultimately, more reliable bottom-up estimates of greenhouse gases will lead to more effective mitigation

strategies.

**Conclusion**

We developed and deployed a multirotor uncrewed aircraft and AirCore system to measure vertical profiles of wind velocity and the mole fractions of $CH_4$ and $CO_2$ downwind of dairy farm operations in the San Joaquin Valley of California. Results from field and laboratory performance evaluations show that the UAAS can reliably measure vertical profiles of wind

velocity and the mole fractions of $CH_4$ and $CO_2$. Integrated with ground-based mobile sampling strategies, UAAS measurement capabilities can increase the vertical resolution of wind velocity and air composition observations in the lower atmosphere, especially in areas where it is difficult to utilize conventional in-situ and remote sensing technologies. Leveraging UAAS and dispersion modeling capabilities can also help detect and quantify greenhouse gas emissions from large area sources. Overall, our findings support further development of UAAS as a low-cost solution to detect and quantify greenhouse gas emissions.

**Data Availability**

Data from field and laboratory experiments are freely available for participants and upon request.



**Competing Interests**

The authors declare that there are no conflicts of interest associated with the research findings presented in this manuscript.

**Acknowledgements**

Funding was provided by the University of California's Lab Fees Research Program, contract number LFR-18-54858.

**Author Contributions**

Zihan Zhu: Conceptualization, Methodology, Validation, Investigation, Data Curation, and Writing - Original Draft; Javier González-Rocha: Conceptualization, Methodology, Validation, Formal Analysis, Visualization, and Writing - Original Draft; Yifan Ding: Investigation, Formal Analysis, Visualization, and Writing - Original Draft; Isis A. Frausto-Vicencio: 400 Conceptualization, Methodology, Validation, Investigation, Data Curation, and Writing - Review and Editing; Sajjan Heerah: Conceptualization, methodology, Investigation, Data Curation, and Writing - Review and Editing; Akula Venkatram: Software, Resources, Supervision, Funding acquisition, and Writing - Review and Editing; Manvendra Dubey: Resources, Supervision, Funding acquisition, and Writing - Review and Editing; Don R. Collins: Conceptualization, Resources, Supervision and Writing - Review and Editing; Francesca Hopkins:, Conceptualization, Resources, Supervision, Project 405 administration, Funding acquisition, and Writing  - Original Draft.

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
