# Peer review of "Toward on-demand measurements of greenhouse gas emissions using an uncrewed aircraft AirCore system"

_EGUsphere, 2023_

## Referee Comment (RC2)

This study evaluates the performance of a novel multirotor unmanned aircraft system for measuring vertical profiles of wind speed, $CH_4$, and $CO_2$ mole fractions. It evaluates $CH_4$ emissions from dairy farm operations using new method, which are crucial for environmental monitoring and understanding the impact of such emissions on the atmosphere. Nevertheless, there are some critical points that require further clarification, such as: the collection and profile retrieval of the AirCore sample are poorly described (or not described at all). This should be clarified and improved, along with the other specific comments mentioned below. Furthermore, additional information about the dairy farm, including details about the farm itself, cattle types, diet, manure management, and more, is necessary. This basic information is essential for readers to assess what emissions could be expected and judge whether the generated estimates make any sense. Also, the $CH_4$ emission estimates from dairy operations need to be expressed using a more suitable unit (see the specific comments for more details) since this study uses short-period measurement.

**Specific comments:**

**L90** Van driving speed expressed in units of km/h, while later in the manuscript, m/s is used for wind speed. Use either km/h or m/s as the unit for velocity, instead of both.

**L98** Ground-based meteorological and **gas sensors** -> Were any gas sensors used alongside the gas analyzer (Picarro G1301)? If so, clarification is needed because gas "sensors" are typically employed for detecting the presence of gases. They are often simpler compared to gas "analyzers", which provide quantitative measurements of multiple gases and are more suitable for research and detailed environmental monitoring applications.

**L114 – L117** Which species were measured using the CRDS analyzer? At what cavity pressure and frequency were the collected samples analyzed? Precision?

**L119 – L124** Mean sUAS speed during the flight? On average, distance of flight tracks compared to the observed source.

**L125 – L142** Lacking a proper description of sample collection and profile retrieval. How were the starting and ending points of the collected sample identified? What is the sampling flow rate of the micro pump attached to the AirCore? The spatial resolution of AirCore measurements?

**L184 – L190** More information on the farm itself, cattle (average weight), milk production, feed management, and the ratio of dry/young to mature (lactating) cattle is necessary to identify if the CH4 emission estimate is reasonable.

**L194** "...$CH_4$ emissions from the **enteric fermentation** or **manure management**..." ->

Enteric fermentation and manure emissions appear here for the first time; this needs to be introduced in the introduction (dedicating a small section to dairy cow emissions and also what has been done until now using different quantification techniques and methods, etc.). It cannot appear out of nowhere in the middle of the manuscript.

**L200** …$C_b$ is the **background CH$_4$** measured from the UAAS… -> How is the CH$_4$ background determined?

**L242** Figure 6. Comparison of UAAS and MET **wind speed** observations. -> not only wind speed, but also the wind direction is presented in Figure 6

**L269** Figure 8. -> y-axis CO2 and CH4 -> $CO_2$, $CH_4$

**L269** Also the description should be clearer "Comparison of UAAS and ground-level CRDS observations **of CH$_4$ and CO$_2$**." …of CH$_4$ and CO$_2$ mole fractions or profiles?

**L306** Figure 9. -> x-axis CO2 and CH4 -> $CO_2$, $CH_4$

**L310** "We selected this set of measurements…" -> Which set of measurements? 20 January 2020? Or all three dates? State it clear.

**L316** "As shown in Figure 10, the dairy farm operation, which is denoted by a **black rectangle**…" -> there is no black rectangle in Figure 10

**L318 – L319** The dairy farm emission estimate represents the whole-farm emission estimate (enteric fermentation + manure emissions) or per animal? Make it clear.

**L320** Indicate the wind direction on the footprint map by adding the arrow that indicates where the wind is coming from.

**L325 – L335** The CH$_4$ emission estimate from dairy operations is presented as Gg yr-1, which is ambitious for short-period measurements of ~11-12 minutes. This appears to be an initial attempt at a new methodology, so the focus should solely be on a critical evaluation of the methodology and emissions over daily or shorter timeframes. Also, a more suitable unit is needed, such as kg/cow(head)/day or kg/AU/day, for comparison purposes with other studies or inventories. Where do the results from your study stand compared with dairy cow farm estimates from other studies/inventories?

---

## Author Comment (AC2)

We sincerely appreciate the reviewer's comments and suggestions. The reviewer's insightful feedback has been very valuable for improving the clarity and presentation of our work. We have carefully considered each comment and suggestion, and have made corresponding revisions to address any critical issue.

**First, when using the UAAS tilt to calculate the wind speed – does the model consider the payload underneath the drone? Since the AirCore and the drone are connected by a 5-meter-long stainless-steel tube, I am wondering whether this part will affect the model results or not. Also, in theory, will this algorithm be accurate at higher wind speed?**

The reviewer's concern regarding the limitations of the kinematic model used to infer wind velocity is well-noted. The kinematic model does not account for the payload carried underneath the hexacopter, likely resulting in wind speed estimation errors as wind conditions increase since the tilt range of aircraft is limited by the added weight. Wind direction estimates obtained from the kinematic model, on the other hand, are not as much affected by the aircraft payload. Future work will explore how higher-fidelity rigid-body models like the ones characterized by Gonzalez-Rocha et al. (2019, 2020), which do account for aircraft mass, can improve the reliability of UAS-based wind estimates.

González-Rocha, J., Woolsey, C.A., Sultan, C. and De Wekker, S.F., 2019. Sensing wind from quadrotor motion. Journal of Guidance, Control, and Dynamics, 42(4), pp.836-852.

González-Rocha, J., De Wekker, S.F., Ross, S.D. and Woolsey, C.A., 2020. Wind profiling in the lower atmosphere from wind-induced perturbations to multirotor UAS. Sensors, 20(5), p.1341.

**Second, it seems that the sample collection and analysis procedure of AirCore is not carefully described. The authors did a flow-through experiment demonstrating the AirCore can preserve CH4 spikes nicely, however, the results of such experiment are not reported in the manuscript. This part is important because air inside AirCore could diffuse during sampling stage & the storage between payload landing & analysis, smearing out peaks & spikes of AirCore samples. Also, will the inside of AirCore release/absorb CO2 and CH4? This can be tested by filling the AirCore with gas of known CO2 and CH4 mixing ratio, then store them overnight before measuring them again (see Karion et al., 2010). Such tests will ensure the quality of AirCore measurement.**

We acknowledge the need for a more detailed account of the AirCore sample collection and analysis procedure. Section 3.3 has been extended to include AirCore characterization results shown below. We also expect the AirCore's Teflon material to minimally influence the release or absorption of CO2 and CH4.

[Figure]

**In addition, the flow pattern during AirCore sampling might need some further clarification – this will be important when registering the CRDS results to altitude. When pumping in air, how does the flow into/out of AirCore look like? Is the pressure gradient inside AirCore in steady state throughout the entire flight? These will all affect the altitude registration of CRDS measurements and can be clarified by reporting results of some simple tests.**

We appreciate the reviewer's input regarding the flow pattern during AirCore sampling. We assumed the pressure gradient is existing to be in steady state and the flow inside the Aircore to be turbulent flow. This article's focus is in combination with the model on interpreting the Aircore's measurements. Detailed discussion about the filling process inside the Aircore can be found in the reference below

Tans, P.: Fill dynamics and sample mixing in the AirCore, Atmos. Meas. Tech., 15, 1903–1916, https://doi.org/10.5194/amt-15-1903-2022, 2022.

**Detailed comments:**

**Line 127: what is the flow rate of micro diaphragm pump when pumping air vs. pumping AirCore? Since AirCore is a long, thin tubing, it may create some resistance to the pump. It is also important to make sure that air is entering the AirCore without too much turbulence. Also, how do you control the on/off of the pump?**

Flow control was achieved by using a metal orifice that effectively constrained the flow rate as long as the upstream vacuum pressure remained below its specific threshold (Refer to the provided flow chart for comprehensive details). Under the vacuum conditions provided by the micro-diaphragm pump at 16" Hg, an inlet flow rate of approximately 0.45 LPM was registered within the Aircore. The operational modulation of the pump was executed by employing a remote relay connected with the pump's power cable.

**Line 135: here the authors introduced the laboratory test of AirCore-CRDS system, however, the results of such tests are not reported in detail. Section 3.3 do not have figures to show the real-time measurements of CH4. In addition, as mentioned above, the "cleanness" of AirCore sampling system need to be carefully checked before measuring real-world samples.**

We thank the reviewer for bringing this information gap to our attention. The real-time CH4 measurements have been visually represented in the figure below. The intended procedure involves preconditioning the Aircore with zero air before starting the sampling process. Unfortunately, due to challenges in preparing zero air source and conducting consecutive measurements, this protocol could not be executed during this deployment. Nevertheless, we ensured that the pump continuously drew in ambient air from the ground for an adequate duration between measurements. Given the generally low ambient concentrations of CH4 and CO2, this approach was expected to yield a consistent and uncontaminated baseline.

[Figure]

**Line 166: in real flights, will the AirCore payload affect the** b3 **Vector?**

Yes, the weight of AirCore is likely to limit how the hexacopter adjusts its attitude in the presence of a wind gust, resulting in a smaller inflow angles and more significant wind speed prediction errors as wind conditions increase. However, the estimates of wind direction obtained by projecting of the b3 vector onto the i1-i2 plane are not as much affected by the weight-induced attenuation of the vehicle's response to wind velocity variations. We have expanded our discussion of the wind estimation results to clarify these two points for the readers.

**Line 223: how long did it take between AirCore landing and analysis during each flight?**

On average, it took less than 5 minutes between AirCore landing and analysis.

**Line 253: how do you define the start of ascent and end of descend? Is there a special gas that distinguish sample air vs. air left inside the AirCore? Will a variable wind speed condition affect your sample collection?**

We placed an ignited lighter in front of the Aircore's inlet before the drone took off. By doing so, a $CO_2$ spike was identified as the start of ascent. The end of descend was identified based on the start of ascend plus the flight time. A variable wind speed condition would not affect the sample collection process.

---

## Author Response (AR1)

We sincerely appreciate the reviewers' comments and suggestions. The reviewers' insightful feedback has been very valuable for improving the clarity and presentation of our work. We have carefully considered each comment and suggestion, and have made corresponding revisions to address any critical issue.

**Reviewer 1 Comments and Response**

**First, when using the UAAS tilt to calculate the wind speed – does the model consider the payload underneath the drone? Since the AirCore and the drone are connected by a 5-meter-long stainless-steel tube, I am wondering whether this part will affect the model results or not. Also, in theory, will this algorithm be accurate at higher wind speed?**

The reviewer's concern regarding the limitations of the kinematic model used to infer wind velocity is well-noted. The kinematic model does not account for the payload carried underneath the hexacopter, likely resulting in wind speed estimation errors as wind conditions increase since the tilt range of aircraft is limited by the added weight. Wind direction estimates obtained from the kinematic model, on the other hand, are not as much affected by the aircraft payload. Future work will explore how higher-fidelity rigid-body models like the ones characterized by Gonzalez-Rocha et al. (2019, 2020), which do account for aircraft mass, can improve the reliability of UAS-based wind estimates. To make this more to the reader we have included the following sentence in Section 4: "more sophisticated dynamic models will be explored as a means to increase the accuracy of wind estimates, both ascending vertically and moving laterally. Previous studies have shown that dynamic models can render higher-fidelity wind estimates from the motion of sUAS (Gonzalez-Rocha, 2019; 2020)."

**Second, it seems that the sample collection and analysis procedure of AirCore is not carefully described. The authors did a flow-through experiment demonstrating the AirCore can preserve CH4 spikes nicely, however, the results of such experiment are not reported in the manuscript. This part is important because air inside AirCore could diffuse during sampling stage & the storage between payload landing & analysis, smearing out peaks & spikes of AirCore samples. Also, will the inside of AirCore release/absorb CO2 and CH4? This can be tested by filling the AirCore with gas of known CO2 and CH4 mixing ratio, then store them overnight before measuring them again (see Karion et al., 2010). Such tests will ensure the quality of AirCore measurement.**

We acknowledge the need for a more detailed account of the AirCore sample collection and analysis procedure. We have expanded Section 2.1. to clarify our data collection procedure by adding the following sentence: "Before each deployment, an ignited lighter was placed in front of the UAAS inlet to mark the starting point of the measurement interval." Additionally, Section 3.3 has been extended to include AirCore characterization results shown below. We also expect the AirCore's Teflon material to minimally influence the release or absorption of $CO_2$ and $CH_4$.

[Figure]

**In addition, the flow pattern during AirCore sampling might need some further clarification – this will be important when registering the CRDS results to altitude. When pumping in air, how does the flow into/out of AirCore look like? Is the pressure gradient inside AirCore in steady state throughout the entire flight? These will all affect the altitude registration of CRDS measurements and can be clarified by reporting results of some simple tests.**

We appreciate the reviewer's input regarding the flow pattern during AirCore sampling. We assumed the pressure gradient is existing to be in steady state and the flow inside the Aircore to be turbulent flow. Detailed discussion about the filling process inside the Aircore can be found in the reference below

Tans, P.: Fill dynamics and sample mixing in the AirCore, Atmos. Meas. Tech., 15, 1903–1916, https://doi.org/10.5194/amt-15-1903-2022, 2022.

**Detailed comments:**

**Line 127: what is the flow rate of micro diaphragm pump when pumping air vs. pumping AirCore? Since AirCore is a long, thin tubing, it may create some resistance to the pump. It is also important to make sure that air is entering the AirCore without too much turbulence. Also, how do you control the on/off of the pump?**

Flow control was achieved by using a metal orifice that effectively constrained the flow rate as long as the upstream vacuum pressure remained below its specific threshold. Under the vacuum conditions provided by the micro-diaphragm pump at 16" Hg, an inlet flow rate of approximately 0.45 LPM was registered within the Aircore. To make this clear we have modified Section 2.4.1 to include the following sentence: "Airflow through the AirCore system was held constant at approximately 0.45 standard liters per minute using an O'Keefe Controls No. 9 (0.02286 cm diameter) metal orifice."

Additionally, the operational modulation of the pump was executed by employing a remote relay connected with the pump's power cable. We have included the following sentence in the Section 2.4.1: "The activation of the AirCore was achieved using a remote relay connected to the pump's power cables."

**Line 135: here the authors introduced the laboratory test of AirCore-CRDS system, however, the results of such tests are not reported in detail. Section 3.3 do not have figures to show the real-time measurements of CH4. In addition, as mentioned above, the "cleanness" of AirCore sampling system need to be carefully checked before measuring real-world samples.**

We thank the reviewer for bringing this information gap to our attention. The real-time $CH_4$ measurements have been visually represented in the figure below. The intended procedure involves preconditioning the Aircore with zero air before starting the sampling process. Unfortunately, due to challenges in preparing zero air source and conducting consecutive measurements, this protocol could not be executed during this deployment. Nevertheless, we ensured that the pump continuously drew in ambient air from the ground for an adequate duration between measurements. Given the generally low ambient concentrations of $CH_4$ and $CO_2$, this approach was expected to yield a consistent and uncontaminated baseline.

[Figure]

**Line 166: in real flights, will the AirCore payload affect the b3 Vector?**

Yes, the weight of AirCore is likely to limit how the hexacopter adjusts its attitude in the presence of a wind gust, resulting in a smaller inflow angles and more significant wind speed prediction errors as wind conditions increase. However, the estimates of wind direction obtained by projecting of the $b_3$ vector onto the $i_1$-$i_2$ plane are not as much affected by the weight-induced attenuation of the vehicle's response to wind velocity variations. We have expanded our discussion of the wind estimation results to clarify these two points for the readers by including the following sentences: "However, a more thorough comparison of UAAS and ground-based wind observations is required to assess if the wind speed errors observed aloft are the result of extrapolation errors associated with the wind profile power law, and to determine the full range of wind conditions for which the wind estimation scheme used to infer wind velocity is reliable. Overall, we expect the wind speed estimation errors to increase as wind conditions intensify since the tilt range of the aircraft is reduced by the added payload weight."

**Line 223: how long did it take between AirCore landing and analysis during each flight?**

On average, it took less than 5 minutes between AirCore landing and analysis. To make this detail clearer for the reader, we have modified Section 2.1 to include the following sentence: "The air sample collected on board the UAAS was analyzed within a 5-minute period upon landing."

**Line 253: how do you define the start of ascent and end of descend? Is there a special gas that distinguish sample air vs. air left inside the AirCore? Will a variable wind speed condition affect your sample collection?**

We placed an ignited lighter in front of the AirCore's inlet before the drone took off. By doing so, a $CO_2$ spike was identified as the start of ascent. The end of descend was identified based on the start of ascend plus the flight time. A variable wind speed condition would not affect the sample collection process. Section 2.1 has been modified to include the following sentence: "Before each deployment, an ignited lighter was placed in front of the UAAS inlet to mark the starting point of the measurement interval."

**Reviewer 2 Comments and Response**

**L90 Van driving speed expressed in units of km/h, while later in the manuscript, m/s is used for wind speed. Use either km/h or m/s as the unit for velocity, instead of both.**

Thank you for the suggestion. We agree that velocity units should be consistent throughout the manuscript. In response, we have modified the text in line 90 as follows: "$CH_4$ and $CO_2$ surveys were first conducted downwind of dairy farm facilities before each deployment (see Figure 1a) by sampling through the inlet of a Picarro G1301 cavity ring-down spectrometer (CRDS) that was placed through the side window of a van driving at a speed of approximately 9 m $s^{-1}$."

**L98 Ground-based meteorological and gas sensors -> Were any gas sensors used alongside the gas analyzer (Picarro G1301)? If so, clarification is needed because gas "sensors" are typically employed for detecting the presence of gases. They are often simpler compared to gas "analyzers", which provide quantitative measurements of multiple gases and are more suitable for research and detailed environmental monitoring applications.**

We agree with the reviewer that the term gas sensor can be misleading. We have modified the title of Section 2.2 to make clear that a gas analyzer instrument was used. Line 98 now reads as follows: "Ground-based meteorological and gas analyzer instruments."

**L114 – L117 Which species were measured using the CRDS analyzer? At what cavity pressure and frequency were the collected samples analyzed? Precision?**

The Picarro G1301 measures $CH_4$, $CO_2$, and $H_2O$ vapor. To make this clearer for the reader, we included the following sentence: "The instrument measures $CH_4$, $CO_2$, and water vapor gas at varying sampling rate ranging between 0.1 and 0.3 Hz." The instrument's tested precision for methane is approximately 10 ppb, which is very small compared to the uncertainty introduced by differences in the AirCore sampling time as shown in Table 3. During de analysis of AirCore samples, the cavity pressure of the Picarro G1301 was approximately 4.5 mbar. The manuscript was modified as follows: The instrument's precision, flow rate, and pressure while conducting $CH_4$ surveys were measured to be 10 ppb, 0.7 standard liters per minute, and 4.5 mbar, respectively.

**L119 – L124 Mean sUAS speed during the flight? On average, distance of flight tracks compared to the observed source.**

During flight operations, the UAAS sampled the air while steadily ascending and descending at an approximate speed of 0.5 m $s^{-1}$. We have modified Section 2.1 to include the following sentence: "The mean speed of the ascent and descent flight operations was approximately 5 m $s^{-1}$." Additionally, we have expanded Table 1 to include the distance from source during each flight operation.

**L125 – L142 Lacking a proper description of sample collection and profile retrieval. How were the starting and ending points of the collected sample identified? What is the sampling flow rate of the micro pump attached to the AirCore? The spatial resolution of AirCore measurements?**

To mark the starting point of each measurement interval, we placed an ignited lighter in front of the AirCore's inlet before taking off. The end point of each measurement interval was determined from the recorded landing time. We have modified Section 2.1 to include the following sentence: "Before each deployment, an ignited lighter was placed in front of the UAAS inlet to mark the starting point of the measurement interval."

Additionally, the AirCore's flowrate was measured to be 0.45 standard liters per minute. We have added following sentence in Section 2.4.1: "Airflow through the AirCore system was held constant at approximately 0.45 standard liters per minute using an O'Keefe Controls No. 9 (0.02286 cm diameter) metal orifice."

Lastly, the spatial resolution was determined to be 5 m based on the characterization results discussed in Section 3.3. We have expanded Section 3.3 to include the following sentence: From these results we conclude that the UAAS has a spatial resolution of 5 m while flying at a steady rate of 0.5 m $s^{-1}$.

**L184 – L190 More information on the farm itself, cattle (average weight), milk production, feed management, and the ratio of dry/young to mature (lactating) cattle is necessary to identify if the CH4 emission estimates is reasonable.**

Thank you for the suggestion. We added Table 3, which describes the herd size and average weights of different animal classes. Information on feed management was not

available in permit data, and goes beyond the scope of this study. See changes to lines 192-194: "The methane emission sources on this dairy farm consists of wet manure management in five manure lagoons, and enteric fermentation from 3115 milk cows and associated support stock housed in three freestall barns and three cattle corrals (Figure 5b; Table 2). Surface area estimates derived from Figure 5b, and estimates of number of animal units derived from permit data (Table 3)."

**L194 "…CH4 emissions from the enteric fermentation, Enteric fermentation and manure emissions appear here for the first time; this needs to be introduced in the introduction (dedicating a small section to dairy cow emissions and also what has been done until now using different quantification techniques and methods, etc.). It cannot appear out of nowhere in the middle of the manuscript.**

We agree with the reviewer that more information on dairy farm methane emissions needs to be provided in the introduction. See additional text on lines 36-37: "Facility-level measurements are particularly needed for dairy farms, which can have a large contribution to $CH_4$ budgets from wet manure management and enteric fermentation emissions, and are important for $CH_4$ mitigation plans in California (Marklein et al. 2021)."

**L200 …Cb is the background CH4 measured from the UAAS… -> How is the CH4 background determined?**

Thank you for pointing out this detail. The lowest mole fraction of $CH_4$ that was measured from the UAAS was used as the $CH_4$ background. We have updated Section 2.6.1 to make this clear by stating the following: "$C_b$ is the lowest mole fraction of $CH_4$ measured from the UAAS."

**L242 Figure 6. Comparison of UAAS and MET wind speed observa5ons. -> not only wind speed, but also the wind direc5on is presented in Figure 6.**

Thank you for the suggestions. The caption of Figure 6 has been modified as follows: "A comparison of UAAS and MET observations of wind speed and wind direction."

**L269 Figure 8. -> y-axis CO2 and CH4 -> CO2, CH4**

$CO_2$ and $CH_4$ abbreviations have been corrected in Figure 8.

**L269 Also the description should be clearer "Comparison of UAAS and ground-level CRDS observations of CH4 and CO2." …of CH4 and CO2mole fractions or profiles?**

Thank you for the suggestion. The Figure 8 caption has been modified as follows: Comparison of UAAS (red) and ground-level (blue) observations of $CH_4$ and $CO_2$.

**L306 Figure 9. -> x-axis CO2 and CH4 -> CO2, CH4**

$CO_2$ and $CH_4$ abbreviations have been corrected in Figure 9.

**L310 "We selected this set of measurements…" -> Which set of measurements? 20 January 2020? Or all three dates? State it clear.**

Thank you for pointing out this ambiguity. We have modified L310 as follows, "The vertical profiles of wind velocity and $CH_4$ that were collected from the UAAS operation performed on January 20[th], 2020 were used as inputs for the dispersion model described in Section 2.6.2 to quantify $CH_4$ emissions from an isolated dairy farm."

**L316 "As shown in Figure 10, the dairy farm operation, which is denoted by a black rectangle…" -> there is no black rectangle in Figure 10**

Thank you for pointing out this error. We have modified L316 as follows: "As shown in Figure 10, the dairy farm operation, which is denoted by a white rectangle…"

**L318 – L319 The dairy farm emission estimate represents the whole-farm emission estimate (enteric fermentation + manure emissions) or per animal? Make it clear.**

Thank you for the suggestion. We re-worded lines 337-338: "Results from this analysis show that whole-farm emissions for this dairy were on average 226 kg $hr^{-1}$ with a lower limit of 140 kg $hr^{-1}$ and an upper limit of 277 kg $hr^{-1}$."

**L320 Indicate the wind direction on the footprint map by adding the arrow that indicates where the wind is coming from.**

Thank you for the suggestion. We have modified Figure 10 to include a wind vector.

**L325 – L335 The CH4 emission estimate from dairy operations is presented as Gg yr-1, which is ambitious for short-period measurements of ~11-12 minutes. This appears to be an initial attempt at a new methodology, so the focus should solely be on a critical evaluation of the methodology and emissions over daily or shorter timeframes. Also, a more suitable unit is needed, such as kg/cow(head)/day or kg/AU/day, for comparison purposes with other studies or inventories. Where do the results from your study stand compared with dairy cow farm estimates from other studies/inventories?**

We used environmental permit data for the studied farm to get an estimated of AU, and converted to units of g/AU/day as in Arndt et al. 2018. Selecting for results from a similar season (winter) and management practice (milk cows housed in freestall barns), we found that our results were comparable with whole-farm emissions estimates from dairy 1 of the Arndt et al. (2018) study, and have added that to the text. Lines 353-357: "we estimated facility emissions of 5430 kg $d^{-1}$ (with a range of 3370-6660 kg $d^{-1}$). This range overlaps with the yearly estimated methane emissions for this particular farm of 3950 kg $d^{-1}$, assuming emissions are evenly spaced over the course of a year, from a model that accounts for the number of cows and manure management practices (Marklein et al.

2021). After normalizing for herd size, our estimated emissions of 714 g/AU/d (range of 444-876) are similar to those measured in wintertime at another California dairy with comparable management practices, 752 g/AU/day (range of 700-803) (Arndt et al. 2018)."

---

## Author Response (AR2)

We sincerely appreciate the reviewer's comments and suggestions. The reviewer's insightful feedback has been very valuable for improving the clarity and presentation of our work. We have carefully considered each comment and suggestion, and have made corresponding revisions to address any critical issue.

1. **In section 2.1, the authors stated, "The mean speed of ascent and descent flight operations was approximately 5 m s-1". I am not sure if this is correct because 5 m/s * 300s = 1500 m. but your profiles only go to ~150 m. Perhaps you meant 0.5 m/s?**

   We thank the reviewer for bringing the ascent rate typo found in Section 2.1 to our attention. The mean speed of ascent and descent was approximately 0.5 m/s. We have modified the sentence in line 104 as follows: "The mean speed of ascent and descent flight operations was approximately 0.5 m s-1"

2. **In figure 10, the wind direction shown on the figure is from north to south, but in the caption, the authors stated the dairy farm is upwind of the observation site. Please address this contradiction.**

   In Figure 10 the wind direction was measured to be from the south while ascending up to approximately 120 m above ground level (blue signal), changing only after the aircraft began to descend (red signal). For this reason, the ascent measurements of wind velocity and air composition were used to estimate an emission flux. During the ascent time the drone was in fact located down wind of the dairy farm shown in Figure 11.

3. **Why only one of the profiles was analyzed using the dispersion model? It would be nice if the authors can report the modeled results of all four profiles, showing how variable such emission can be, and what are the major sources of uncertainty.**

   We agree with the reviewer that more analysis is needed determine the variability of emission sources. However, we were unable to determine reliable background measurements during flights 2, 3, and 4. Furthermore, flights 2 and 3 were conducted near dairy farm clusters whose management practices may be different, making it difficult to disentangle emission sources. For these reasons, our manuscript presents dispersion modeling results based on wind velocity and mole fraction measurements collected during the first flight.

---

## Author Response (AR3)

Copyright symbols have been added to Figures 1 and 5.